# Learning massive interpretable gene regulatory networks of the human brain by merging Bayesian networks

**Niko Bernaola**[1], **Mario Michiels**[2], **Pedro Larrañaga**[1]*, **Concha Bielza**[1]

**1** Computational Intelligence Group, Departamento de Inteligencia Artificial, Universidad Politécnica de Madrid, Madrid, Spain, **2** Centro Integral de Neurociencias Abarca Campal, Hospital Universitario HM Puerta del Sur, Madrid, Spain

* pedro.larranaga@fi.upm.es

## Abstract

We present the Fast Greedy Equivalence Search (FGES)-Merge, a new method for learning the structure of gene regulatory networks via merging locally learned Bayesian networks, based on the fast greedy equivalent search algorithm. The method is competitive with the state of the art in terms of the Matthews correlation coefficient, which takes into account both precision and recall, while also improving upon it in terms of speed, scaling up to tens of thousands of variables and being able to use empirical knowledge about the topological structure of gene regulatory networks. To showcase the ability of our method to scale to massive networks, we apply it to learning the gene regulatory network for the full human genome using data from samples of different brain structures (from the Allen Human Brain Atlas). Furthermore, this Bayesian network model should predict interactions between genes in a way that is clear to experts, following the current trends in explainable artificial intelligence. To achieve this, we also present a new open-access visualization tool that facilitates the exploration of massive networks and can aid in finding nodes of interest for experimental tests.

## Author summary

In this study, we have developed a faster and scalable method, the Fast Greedy Equivalence Search (FGES)-Merge, to understand how genes interact and regulate each other. We adapted it specifically for massive gene regulatory networks, which can have tens of thousands of genes. Our method is not only competitive with the current best methods in terms of accuracy but also outperforms them in terms of speed. This is crucial when working with large scale data such as the human genome.

To make our findings clear and usable for fellow scientists, we also created an open-access visualization tool. This tool allows for exploring vast networks and identifying nodes of interest for further research. In our test cases, we used the FGES-Merge method to learn about the gene regulatory network of the entire human brain, using data from various brain structures.

**Data Availability Statement:** All data, models used for testing and generating figures and code for the algorithm are available at https://gitlab.com/mmichiels/fges_parallel_production/-/tree/master

The original dataset is available from the Allen Human Brain Atlas at https://human.brain-map.org.

**Funding:** This project has received funding from the European Union's Horizon 2020 Framework Programme for Research and Innovation under Specific Grant Agreement No. 785907 to PL (HBP SGA2), and Specific Grant Agreement No. 945539 to CB (HBP SGA 3). The project has also received funding through the Spanish Ministry of Science and Innovation through the projects PID2022-139977NB-I00 to CB and TED2021-131310B-I00 "Bayesian Networks for Interpretable Machine Learning and Optimization" (BAYES-INTERPRET) to PL. The founders did not play any role on study design, data collection, analysis, decision to publish or preparation of the manuscript.

**Competing interests:** The authors have declared that no competing interests exist.

Our work provides a significant step towards accurately predicting gene interactions on a large scale and more quickly than before. This can guide future biological research by letting scientists test the interactions our method predicts, thereby furthering our collective understanding of gene functions.

## Introduction

With the advent of high-throughput measurement technologies in biology in the 1990s, such as in situ hybridization [1, 2] or RNA microarrays [3], it has been possible to collect information for tens of thousands of genes from every tissue sample. By studying the level of expression of each gene in different conditions it is possible to reconstruct the underlying regulatory relationships between the genes and, therefore, get closer to understanding their function [4, 5]. Due to the combinatorial nature of gene regulation [6] and the size of the genome (which can have tens of thousands of genes), it would be intractable to experimentally determine all of the regulatory links. Even further, a complete model would have to take into account post-transcriptional modifications. To solve this problem, many computational methods have been proposed to infer the gene regulatory network (GRN) from expression data [7, 8]. The models learned can then be used to guide biological research by letting researchers test the interactions predicted by the network. More generally, we are interested in finding solutions to massively high-dimensional problem while maintaining the ability to interpret the results. Following [9], on which we base our method, "Ideally, one should be able to analyze tens of thousands of variables on a laptop in a work session and be able to analyze problems with a million variables or more on a super-computer overnight". For this reason, we decided to use Bayesian networks (BNs) [10, 11] with our main contributions being presenting an algorithm that scales to tens of thousands of variables, despite the known intractability of structure learning in general BNs, by being adapted specifically for the topology of GRNs and a visualization tool that helps experts understand the network despite its massive size.

In the next section, we introduce BNs, their properties and how their structure can be learnt from data. We then present the biological background required to understand GRNs and how the properties of GRNs restrict the space of possible BN structures that model them. Afterwards, we go into the detailed steps of our algorithm and BayeSuites, the tool developed for visualizing the resulting massive BNs. Then, we show the results of applying our method to the DREAM5 Network Inference benchmark, which uses real and simulated data with a gold standard network to evaluate the effectiveness and scalability of our method, and, to showcase the ability to scale to even larger networks, the Allen Human Brain Atlas dataset, including the visualization of the reconstructed networks using FGES-Merge and BayeSuites. Finally, we discuss the assumptions required for our algorithm to work and the limitations of our method and our results. Acknowledging that the objective of learning a model of all gene interactions is very far from done, we end by presenting some avenues for further work improving our method to relax some of the assumptions and overcome some of the limitations.

### Theoretical background

In this section, we present the theoretical background required to understand BNs: how to learn the structure from data, how to learn the parameters of the associated probability distribution and how to use the model to understand the dependencies between nodes. Then, we introduce the biological background of the problem of reconstructing GRNs from data using

BNs and explain how the concrete characteristics of the problem can be used to constrain the possible networks and how BNs can help with the problem of understanding gene regulation.

**Bayesian networks.**   BNs [10] are probabilistic graphical models that combine probability and graph theory to efficiently represent the probability distribution of a group of variables $\mathcal{X} = \{X_1, \ldots, X_n\}$. For the rest of this section, unless stated otherwise, we follow [12] for definitions and equations. BNs model probabilistic conditional dependencies and independencies between the variables in $\mathcal{X}$ in terms of:

- A directed acyclic graph (DAG), $\mathcal{G}$. $\mathcal{G}$ is a pair $(V, E)$, where $V$ is the finite set of vertices or nodes indexed by $\mathcal{I}$ and $E$ is a subset of $\mathcal{I} \times \mathcal{I}$, with element $(i, j)$ indicating an edge between nodes $i$ and $j$. Each of the nodes in the graph denotes a variable in $\mathcal{X}$ with the edges representing conditional probabilistic dependencies between the variables

- A series of conditional probability distributions (CPDs). Each of the CPDs is associated with a variable $X_i$ and gives the probability distribution of that variable conditioned on its parents in the graph, that is, the nodes that have edges directed towards $X_i$, which we denote $\mathbf{Pa}(X_i)$. The formula for the joint probability distribution of the variables given all the CPDs is:

$$p(X_1, \ldots, X_n) = \prod_{i=1}^{n} p(X_i | \mathbf{Pa}(X_i)) \tag{1}$$

We are interested in problems where we have a dataset $\mathcal{D} = \{\mathbf{x}^{(1)}, \ldots, \mathbf{x}^{(N)}\}$ where $\mathbf{x}^{(i)} = (x_1^{(i)}, \ldots, x_n^{(i)}) \in \mathbb{R}^n, i = 1, \ldots, N$, with $N$ the number of measurements of the $\mathcal{X}$ variables. We assume the entries of $\mathcal{D}$ are i.i.d. samples of a joint probability distribution and we will use them to find a BN that represents the underlying distribution of the data. We can separate the problem of finding the BN in two parts: finding the structure of $\mathcal{G}$ and finding the CPDs associated to each of the nodes. Given the graph structure, finding the CPDs can be done in polynomial time through maximum likelihood estimation (see chapter 3 of [12]). However, finding the structure of the graph is known to be NP-hard, with the search space growing super-exponentially with the number of variables [13]. Finally, once we have the BN we are interested in using it to run inference. This usually means we care about the question: given that a subset of the variables $\mathbf{E} \subset \mathcal{X}$, that we will call evidence, has taken on some concrete values; what is the new distribution of some (or all) variables of interest in $\mathcal{X} \backslash \mathbf{E}$? This problem has also been shown to be NP-hard in general [14]. Throughout this work we will use Gaussian BNs and we will focus on the problem of learning the structure of the graph.

**Gaussian Bayesian networks.**   Gaussian BNs are a type of BN where all variables have conditional distributions that are Gaussian distributions. A multivariate vector $\mathbf{X} \in \mathbb{R}^n$ is distributed like a Gaussian distribution $\mathcal{N}(\boldsymbol{\mu}; \boldsymbol{\Sigma})$ if its probability distribution is given by:

$$p(\mathbf{X} | \boldsymbol{\mu}, \boldsymbol{\Sigma}) = \frac{1}{\sqrt{2\pi|\boldsymbol{\Sigma}|}} \exp\left\{ -\frac{1}{2}(\mathbf{X} - \boldsymbol{\mu})^T \boldsymbol{\Sigma}^{-1}(\mathbf{X} - \boldsymbol{\mu}) \right\} \tag{2}$$

In a Gaussian BN, each variable $X_i$ with parent variables $\mathbf{Pa}(X_i)$ has a conditional distribution that is Gaussian with a mean equal to a linear combination of the parent variables plus a constant offset:

$$p(X_i | \mathbf{Pa}(X_i)) = \mathcal{N}(\beta_0 + \boldsymbol{\beta}^T \mathbf{Pa}(X_i); \sigma_i^2) \tag{3}$$

The joint distribution over $X_i$ and $\mathbf{Pa}(X_i) = \{X_{i_1}, \ldots, X_{i_k}\}$ is a normal distribution where the

covariance of $X_i$ and each $X_{i_j}, j = 1, \ldots, k$, is:

$$\text{Cov}[X_i; X_{i_j}] = \sum_{j=1}^{k} \beta_{i_j} \Sigma_{i,i_j}, \tag{4}$$

where $\beta_{i_j}$ is the coefficient of the parent $X_{i_j}$ of $X_i$ in the conditional mean of Eq 3.

By applying this conditional Gaussian structure repeatedly, ordering the variables topologically from parents to children, we can show that the joint distribution over all variables in a Gaussian BN is also Gaussian. The mean vector and covariance matrix of the joint distribution can be derived by iteratively combining the conditional distributions.

The key benefit of Gaussian BNs is that many calculations involving Gaussians can be done in closed form. This makes inference and parameter learning more efficient than in networks with non-Gaussian distributions. However, Gaussians may not be able to capture non-linear effects and dependencies that other distributions could represent.

**Parameter learning.**   To learn the parameters of a Gaussian BN, we can fit a regression for each variable $X_i$ with its parent variables $\mathbf{Pa}(X_i)$ as predictors. The coefficients of this regression give estimates of the mean parameters $\beta_0$ and $\boldsymbol{\beta}$ of the conditional distribution of $X_i$ given $\mathbf{Pa}(X_i)$ (Eq 3). The mean squared error of the regression gives an estimate of the variance parameter $\sigma_i^2$. By fitting these regressions for each variable in the network, we can estimate all the parameters of the Gaussian BN.

**Inference.**   Exact inference in a Gausian BN is much more feasible than it would be for a general BN. For conditioning, we follow a procedure which requires converting the Gaussian into information form (for which we need to invert the covariance matrix) and setting the values of the evidence variables. We do not need to invert the whole matrix, just the block containing the evidence variables. To obtain the posterior CPD for variable $X$ (we omit the subscript for an easy notation) after conditioning on a set of evidence variables $\mathbf{E}$, we use Eqs 5 to 8:

$$p(X|\mathbf{E}) = \mathcal{N}(\tilde{\beta}_0 + \tilde{\boldsymbol{\beta}}^T \mathbf{E}; \sigma_X^2) \tag{5}$$

$$\tilde{\beta}_0 = \mu_X - \Sigma_{X\mathbf{E}} \Sigma_{\mathbf{EE}}^{-1} \boldsymbol{\mu}_{\mathbf{E}} \tag{6}$$

$$\tilde{\boldsymbol{\beta}} = \Sigma_{\mathbf{EE}}^{-1} \Sigma_{X\mathbf{E}} \tag{7}$$

$$\tilde{\sigma}_X^2 = \Sigma_{XX} - \Sigma_{X\mathbf{E}} \Sigma_{\mathbf{EE}}^{-1} \Sigma_{\mathbf{E}X} \tag{8}$$

$\Sigma_{X\mathbf{E}}$ refers to the block of the covariance matrix relating the evidence variables and $X$. $\Sigma_{\mathbf{EE}}$ refers to the covariances of the evidence variables and it is the only matrix that has to be inverted. $\tilde{\beta}_0, \tilde{\boldsymbol{\beta}}$ and $\tilde{\sigma}_X$ give the new mean and standard deviation of $X$ after conditioning on the evidence $\mathbf{E}$.

Normally, inference in BNs is exponential in the number of nodes, but for Gaussian BNs, it is reduced to matrix multiplication and inversion so the complexity is $O(l^3)$ where $l = \max(|\mathbf{E}|, n - |\mathbf{E}|)$, the bigger of the two matrices we have to invert and multiply. This makes it tractable even for large networks.

**Structure learning.**   The process of structure learning involves identifying the optimal DAG that represents the dependencies between variables in the dataset. We will only present the required background to understand our contributions. For a recent in-depth review of structure learning methods for BNs see [15].

There are two main approaches for learning the structure of a BN: Through expert elicitation of the dependencies between variables or directly from data. As we mentioned in the introduction, it would be intractable for such high-dimensional problems to defer only to expert knowledge when building the network, so we need to use data-based methods. However, we mention expert knowledge because it can be a way to either restrict the search space by adding partial information about the structure, or a way to check the accuracy of data-based methods by comparing their predictions with expert consensus.

Of the data-based methods, we will focus on score-based methods which search through the space of possible network structures and select the one that optimizes a given scoring function. The scoring function measures the goodness-of-fit of the structure to the data. For example, the score we will use is a modified version of the Bayesian Information Criterion (BIC) and is given by:

$$\text{BIC}(\mathcal{D}|\mathcal{G}) = \log L(\mathcal{D}|\mathcal{G}) - \lambda d \log N \tag{9}$$

where $L(\mathcal{D}|\mathcal{G})$ is the log-likelihood of the data given the model, $d$ is the number of parameters, $N$ the number of samples and $\lambda$ a hyperparameter of the model to adjust how much we penalize complexity. More concretely, for a Gaussian BN, the BIC of the linear Gaussian at each node $X_i$ is the sum of the residuals of the regression with $\mathbf{Pa}(X_i)$. More importantly, it has been shown that under some conditions, searching the space by adding the edge that increases the BIC the most starting from an empty model guarantees that asymptotically those models will have the highest posterior probability given the data.

## Gene regulatory networks

The processes that regulate which genes are expressed and how much of the protein is synthesized are the topic of study of genetic regulation. In general, we are interested in seeing how different environmental conditions (internal or external) change the amount by which each protein is synthesized. However, biological data is noisy and the processes it studies are stochastic, so any relationship we find will have to be probabilistic.

One common way of simplifying the problem is to assume that all regulations can be modelled as genetic interactions [7]. This means that we assume that any non-genetic factor (hormones, temperature, chemical changes in the environment, etc.) will not have an effect on the level of expression of genes except indirectly, via mediation by another gene. This network structure of interactions is the GRN. In this work, we will make the additional, very common assumption [5, 16], of taking the steady-state expression level which means we will work with static instead of dynamic models.

**Data collection.**   As is commonly done we use microarrays to obtain gene expression data [17]. This works by lining a chip with microprobes that puncture a biological sample and take a sample of the cytoplasm. Each probe in the array is lined with the complementary chain to the RNA we want to detect (a different one in each probe) in such a way that after washing out the array, only the bound RNA will be found in each probe. The bonding strands are made to induce a fluorescent molecule to emit light when bound so that each probe will emit light corresponding to the amount of binding RNA found in the sample. The main limitations of microarrays are that we can only measure known transcripts (since the complementary strand has to be designed onto the chip) and that the microarrays give a measurement of the population of cells in the tissue sample. Since there might be many different cell types in the population, the measurement may not be representative of any single one of them. For our work, we use the microarray data from the Allen Human Brain Atlas dataset [18].

**GRNs and their properties.**   We are interested in both the topology of the network to see the interactions between genes and in accurately predicting changes in the level of expression of some genes given other changes in the level of other genes. In this section, we will briefly discuss how BNs are a good fit to model the problem of learning the structure of GRNs and some of the most important biological properties that can be used to constrain the space of possible structures. A GRN is defined by a directed graph $\mathcal{G}(V, E)$, and a set of functions for each gene that relate the expression of that gene to that of the other members of the network. In the context of GRNs, the set of nodes is always associated one-to-one with the set of genes. Edges are interpreted as relationships between genes, but the precise definition of the relationship will depend on the mathematical model being used. In our case, $\mathcal{G}$ will be represented by the structure of a BN, the functions related to each gene will be the CPDs of each node and the edges represent conditional probabilities between the expresion of the genes.

**Topological properties of GRNs.**   The topological structure of the network is useful by itself, even if we don't know the functions related to each of the genes as long as it has a proper interpretation. In the case of BNs, two nodes that are disconnected are conditionally independent of each other. This is a probabilistic and not a causal relationship, but it can give a visual intuition of the interactions at play. Some of the most important information we can extract from the topology of the network is the degree distribution, which is given by the number of edges attached to each node of the network. We can further distinguish between the in-degree and out-degree, which relate to the number of edges going into or out of each node, respectively. We use this information to restrict the possible structures of the network we learn since it should follow the correct distributions as explained below. For a more in-depth overview of the topological properties of GRNs see [6, 19, 20].

We will summarize these properties here:

- GRNs are locally dense but globally sparse: GRNs have a small number of edges compared with a fully connected network. The maximum number of edges would be $n^2$ (if we allow for self-edges). The actual number of edges in a GRN varies depending on the network but is typically $O(n)$ with a small constant. In the literature this is referred to as the network being sparse. We remark that this property is due to the total number of edges compared to the number of nodes and will refer to it as the property of global sparsity. However the edges are not uniformly distributed. We do not see each node having a similar number of edges as we would expect if it were. Instead, we see many genes with no edges and some nodes with many edges. This means that the global sparsity of the network does not imply that each of the nodes is sparsely connected. That is: in the case of GRNs, global sparsity does not imply local sparsity. Learning methods that require sparsity, such as [9], assume that the degree distribution is uniform since the way the property is used in the algorithm is to restrict the number of possible combinations of edges for a given node. The number of combinations scales exponentially with the number of edges so if the node is forced to be sparse, we can guarantee that the number of combinations is bounded by a small number.

- The interpretation of the in-degree of a node in a GRN is the number of regulators a gene has. This number is usually small since most genes require just one regulatory factor, although it can be higher in more complex processes that need multiple conditions to be met at the same time. However, there is a physical restriction on the number of regulators. In the case of transcription factors, the only way they can affect the expression of a gene is to be physically close to that gene, affecting the DNA directly. Since there is limited space around each gene, there can only be a limited number of transcription factors and thus a limited number of regulators. This translates to an upper bound to the number of parents, which we will call $s_{max}$ that is small compared with $n$. The in-degree distribution (with values $m = 0, 1,$

...) can be approximated by a power-law distribution up to an upper limit and with an atom at $m = 0$ to account for the possibility of unregulated genes. The distribution is given by:

$$p(m) = \begin{cases} C_0 & \text{if } m = 0 \\ Cm^{-\alpha} & \text{if } 1 \leq m \leq s_{\max} \\ 0 & \text{otherwise} \end{cases} \qquad (10)$$

where $C_0$ and $C$ will depend on the specific network and represent the fraction of genes that have 0 and 1 regulators, respectively. $\alpha$ is also network dependent and represents how fast the probability decreases as $m$ increases. That GRNs follow this distribution is one of the most important assumptions of our work and we will use it heavily to guide our structure learning algorithm.

- The out-degree of a node in a GRN is the number of genes regulated by it. The distribution is similarly approximately distributed as a power-law. However, although we don't expect any gene to be connected to all other, the upper bound for the out-degree is much looser. The nodes at the tail of the out-degree distribution are hubs of the network, and the biological interpretation is that they are regulators of transcription (transcription factors) that regulate complex processes involving many genes or multiple different processes. In practice, this means that there will be a few nodes with many children, and any node with a degree higher than $s_{max}$ is more likely to be a hub, so most of its edges should be directed outwards from it.

**Modelling GRNs with BNs.** In the field of GRNs, one of the earliest approaches was [11] which used a simple procedure of searching the entire space of structures for the one with the maximum likelihood. This method, while straightforward, can be computationally expensive, especially for large networks since, as we explained before, the space of structures scales super-exponentially with the number of nodes. In [21], the authors used the PC algorithm, which is a constraint-based method that relies on conditional independence tests, on microarray data to obtain a GRN. This method has similar problems of scaling poorly since as the number of nodes increases, the number of tests does too with the number of permutations of possible parents. Works such as [22] circumvent the problem by using a parallelized method implemented in a supercomputer to be able to scale to genome-wide networks.

From very early, work has focused on improving the efficiency of structure learning by restricting the search space instead of increasing computing power. In [23], the authors applied a variant of the Chow-Liu algorithm to learn a BN in quadratic time, but with a severe limitation on the structure since it must be a polytree, while works such as [24] attempted to simplify the problem of learning the structure by including expert knowledge in the form of a prior for the structure. Similarly, in [20], the authors used topological information to restrict the space of structures and accelerate the search. More recently, [25] reviews different ways of introducing prior knowledge as structural restrictions and shows that these restrictions lead to improved results on the reconstruction of the networks.

Another avenue of research has been using a divide-and-conquer approach to reduce the size of the search space. In [26], co-regulated modules of genes were first identified and then a BN was built for each of them. Another approach is [27], in which the authors avoid relying on previous knowledge by learning a small local network around each node to combine them later.

Finally, to reduce the errors in the learnt structure, there have been some approaches using ensemble methods that combine multiple structures into one. For example, [28–30], from the

Network Inference Challenge use bootstrapping to learn multiple networks from the same population and combine them to produce a consensus. Also in the challenge, the authors of [31], use the Catnet R [32] package to learn multiple structures using simulated annealing to introduce variance in the searching procedure and then aggregate the results.

## Design and implementation

### FGES-Merge

FGES-Merge is our proposal for an algorithm capable of efficiently learning massive BNs without forcing initial structural restrictions. We broadly follow the structure of the algorithm from [27] but introduce several improvements that make our algorithm scale from the 1000-node networks tested in the reference algorithm to networks that are 20 times larger, such as the human genome network.

The structure of the algorithm is as follows: First, for each gene $X_i$, we select its most likely neighbours as candidates for a local subgraph around $X_i$. Next, we learn each local subgraph using a modified version of FGES. Finally, we merge the local subgraphs by performing graph unions with pruning to satisfy the topological properties of GRNs.

**Neighbourhood selection.**   We want to simplify the problem of learning a massive BN by dividing the network into a smaller neighbourhood network for each of the nodes and then merging them. To do this, we need to select which nodes will belong to each of the smaller networks. In [27], the authors calculate the pairwise mutual information (MI) between the nodes. Then, for each node $X_i$, they assume the MIs come from two different distributions: one with the nodes in the neighbourhood of $X_i$, and the other with the nodes that are not in the neighbourhood of $X_i$. They assume that any MI sampled from the first distribution will always be higher than any MI from the second. This means that they can sort the MIs from highest to lowest and test the likelihood, for each possible size $s$ of the network, that the first $s$ nodes belong to a distribution and all the others belong to the second distribution and compare it with the likelihood that they are all sampled from just one distribution. Then, they take the most likely neighbourhood size $s_l$ and return the first $s_l$ nodes sorted by their MI with $X_i$ as the candidates for the first neighbourhood network. They then repeat this process for each of the nodes in the original graph (Fig 1).

We modify this procedure by changing our score from the MI of $X_i$ and $X_j$ to the BIC difference [33] of adding the new edge from $X_j$ to $X_i$ (although this will be symmetrical at this stage). This is done because calculating BICs is one of the most expensive steps in the algorithm and we will also need them for the next step. As we can see in Eq 11, the BIC is related to the residual of a linear regression between the nodes and is a way to measure the strength of how strongly related two nodes are. MI has the advantage of being able to also capture non-linear relationships in the data. However, since our model assumes linearity, we would not gain anything from using it. Since the size of the BIC matrix is of order $n^2$ and the calculations are independent of each other we parallelized this step. At this initial step, the set of parents is empty and so the calculation of the BIC just takes into account the variance of the samples and is given by:

$$\mathrm{BIC}(X_i|X_j) = -N\ln(\sigma_i^2) - \lambda k \ln(N) \tag{11}$$

Here $k$ is the number of parents of $X_i$, which would be one in this initial step. Our second modification is limiting the possible size of the neighbourhood to $s_{max}$. We decided to do this because the topological properties of the GRN imply that the set of parents of each node is small. Even though the set of children can be very large for the hubs of the network, we assume that each child will contain the hub in its neighbourhood. This means that when we merge the

**Neighborhood selection**

**Fig 1. Diagram showing the neighbourhood selection step of the FGES-Merge algorithm.** First the pairwise BIC matrix is computed by calculating the BIC score of adding the edge between each pair of nodes, such that every $a_{ij}$ corresponds to $\mathrm{BIC}(X_i|X_j) = -N\ln(\sigma_i^2) - \lambda k\ln(N)$. Then, the BICs are filtered to only take into account the positive values and sorted from highest to lowest. We divide each row at the most likely point and take the left side as the neighbours for the next step.

edges between the hub and its children, we always expect the edge between the hub and each child to be added to the final network. This limitation makes each of the neighbourhood networks smaller thus speeding up the algorithm.

## Fast greedy equivalence search

In this step [27] uses a simple greedy strategy to learn the neighbourhood networks. Instead, we use our own variant of the FGES [9] algorithm which is much faster. FGES starts by greedily searching over the space of edge additions in the forward equivalence search (FES) step. At each step, it adds the best possible edge to the structure of the BN according to the BIC difference from adding that edge, which, if we want to add an edge from $Z$ to $Y$, is given by:

$$\Delta\mathrm{BIC}(Y, Z) = \mathrm{BIC}(Y|\mathbf{Pa}(Y) \cup Z) - \mathrm{BIC}(Y|\mathbf{Pa}(Y)) \tag{12}$$

where each of the BICs is calculated using Eq 11 after learning the multilinear regression of $Y$ against the set of its parents (with and without $Z$). Then, instead of calculating the values of all possible edge additions again, our variant FGES algorithm uses the fact that since the BIC is a local score, the new edge can only modify the score of some of the edges around it. This allows us to skip many of the computations. Once no edge additions are possible (because they would all worsen the graph), we search the space of edge deletions in the backward equivalence search (BES) step to end up with the best scoring structure (Fig 2). The original algorithm is implemented in TETRAD [34].

The main modifications we do are parallelizing the calculation of the new possible edge additions at each step and adding simulated annealing to choose which edges to add or remove. The first change makes the slowest step of the original FGES algorithm much faster. We first create the set of all edges that have changed with the last addition and then divide it across all processors so that we can calculate their scores in parallel.

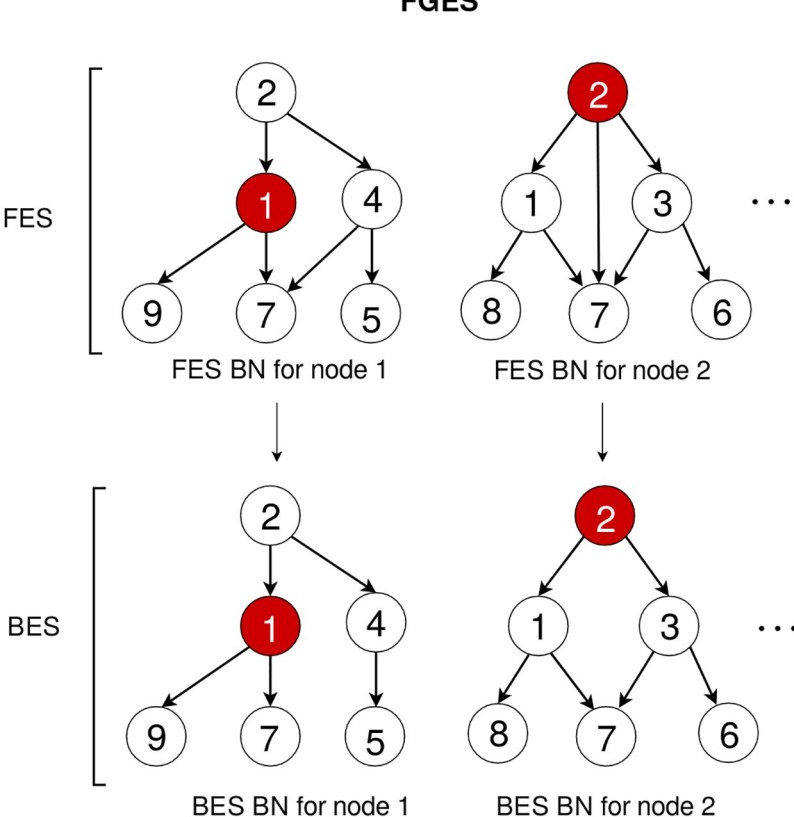

**Fig 2. Basic flow of the FGES algorithm, following the example of Fig 1.** First, we take the neighbourhood candidates from the previous step and greedily (except for the simulated annealing step) add edges until the BIC cannot improve in the FES. Then, we perform any edge deletions that improve the score in the BES. This approach gives us the globally optimal neighbourhood network for the candidates. We then repeat the process for each of the other nodes.

The second change sometimes chooses suboptimal additions and deletions to increase exploration by including a probability that a random addition or deletion will be chosen instead of the maximum scoring one. This probability will decrease with each iteration until it reaches zero, so the final steps of each network are always performed greedily. Given two neighbouring nodes $X_i$ and $X_j$ (in the sense of belonging to the same neighbourhood graph), we are usually going to find them together in many of the small networks. That is, if one of them is selected, the other is very likely to be selected too. By allowing for suboptimal edges, we increase exploration in each of the subgraphs and make it more likely that the final structure contains the true edges, since the best scoring ones in each subnet do not necessarily correspond to the edges in the true network because each subnetwork is missing context. Since after merging we can only eliminate edges, not add new ones, we increase exploration now at the cost of false positives and we rely on the pruning phase to eliminate edges that have low scores in the final graph.

**Merge.** The final step is combining all the learned local networks into a single global network for all the nodes. In [27], the authors try different methods for doing this and conclude that the best scoring method is to simply perform the union of all the graphs while checking for and removing any cycles. We found an improvement to this method by using a pruning

strategy that removes the lowest scoring edges according to their BIC and the number of times they appear in the subgraphs. We sort them by their score and remove the worst performing ones to keep only the expected number of edges in the network, $\mu n$, where $\mu$ is a hyperparameter of the algorithm that should be set to a small constant (approx. between 1 and 10) to be consistent with the expected topology of the network. Since we only calculate the BICs once, at the beginning of the merging phase, this is an approximate strategy since we do not calculate the effect of each removal again. We merge the graphs sequentially so that cycles can only appear in the neighbourhood of the most recently added subgraph. Since this neighbourhood is small the cycle removal step can be done in reasonable time.

We also added a step that allows the user to optionally introduce a predefined list of hubs (which in the case of GRNs would usually correspond to transcription factors) so that edge orientation can be consistent with expert information. If this list is missing the algorithm chooses the nodes with the highest number of neighbours as hubs and, if we find inconsistent orientations during merging, makes the hubs the parents. Again, we found that this change improved the accuracy of edge orientation and made the topological properties of the BN more closely match those of a true GRN. See Fig 3 which summarizes the merging and pruning process.

## Visualization of massive GRNs

Genome-wide GRNs consist of thousands of nodes and edges, making them difficult to visualize. Nevertheless, visualization is crucial for understanding and analyzing these networks. To develop an effective visualization tool for massive networks, two main challenges need to be addressed: computational efficiency and usability.

A common approach to reduce computational burden and make the network tractable for common tools for the analysis of BNs is to show only subgraphs instead of the entire network. While this approach is valid for showing some relevant genes and their connections, we would prefer to have the ability to visualize and work with the whole network at once.

**Developing a web application for BN learning and visualization.**   Our goal was to create a comprehensive modern solution for BN learning and visualization. We present BayeSuites

## Combination

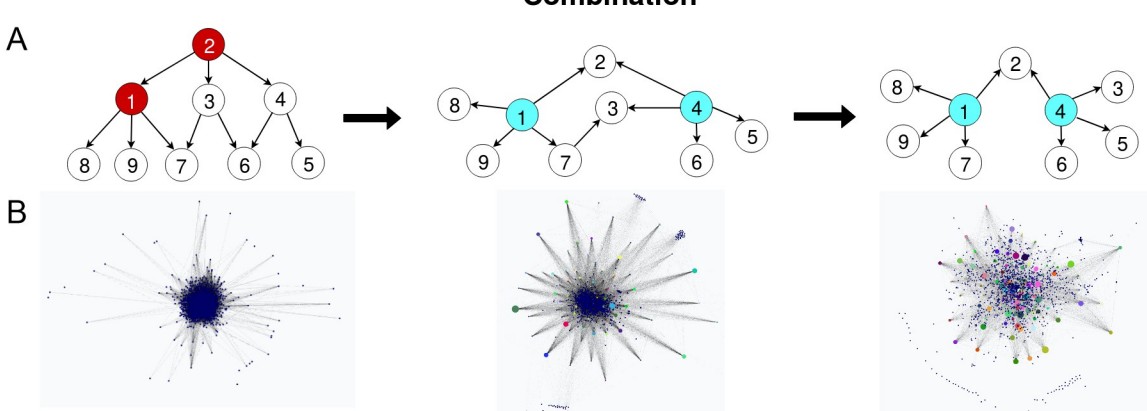

**Fig 3. Diagram explaining the combination steps. A**. Simplified example continuing from Fig 2. First (left), we merge the neighbourhood networks with their union, removing any existing cycles. In the second step (middle), we find the hubs (in blue) either via expert knowledge or by using the most-connected nodes in the network to orient the edges from the hubs to their connected nodes that are not also hubs while checking for cycles and removing them. Finally (right), in the third step we prune the worst performing edges by their BIC score up to a size threshold corresponding to the expected number of edges in a GRN (approximately #edges < 10$n$, see [6]). **B**. Real world use case of the combination procedure with our learned human brain network.

[35], a web application that includes all the desired functionalities, offering ease of use, interactive capabilities and computational efficiency. The web application's software architecture makes it more accessible than desktop software, eliminating the need for multiple package installations or different dependencies for various operating systems and hardware architectures.

We designed the user interface specifically to manage massive networks, resulting in well-polished interactive tools for working with BNs. The whole web service was optimized for visualization, including a separation of the visualization code from the business logic code for managing the graph algorithms (such as computing different layouts).

**Sigma library for graph visualization.**   We used the Sigma library for the graph visualization task. Sigma is a JavaScript library that provides a WebGL backend to leverage GPU resources. GPUs have substantially increased in power in recent years and are now able to solve the problem of visualizing a massive network. Since the library is a general-purpose graph visualization package, we implemented all the necessary modifications to adapt it for BNs.

For proper visualization of BNs, we need to visualize the CPDs, run inference-related operations such as making queries and observing the posterior distributions, implement specific highlighting tools such as showing the Markov blanket of a node, etc. In summary, we need a rich set of interactive tools to fully understand the BN structure and parameters. This is where current BN visualization software frameworks fail for massive BNs, as their implementations of these operations are not scalable to tens of thousands of nodes.

**Interactive tools for BN visualization.**   Now, we will focus on the interactive tools we developed to be able to understand massive BNs. Some of these tools are general purpose graph tools, while others are specific for BNs. One important general purpose tool is the selection of layouts, wich helps position the nodes and edges in a meaningful way. Though every layout can be used for BNs, force-directed layouts such as the Fruchterman-Reingold or ForceAtlas2 algorithms [36] are recommended for GRNs, as they help identify hubs, connected components, and disconnected genes (see Fig 4 to view a selection of the available layouts).

To understand massive networks, we usually need multiple ways to find and select the desired nodes. To highlight important nodes, we offer two main options: a user defined list of nodes ordered by groups or a set of automatic detection algorithms based on topological properties or community grouping [37]. Usually, the combination of a force-directed layout with the Louvain algorithm provides interesting insights into possible clusters in the network structure. Force-directed algorithms separate disconnected components and cluster connected nodes around hubs, that, when coloured, give an initial idea of what the components of the network might be.

To provide a use case for dealing with a user-defined list of groups, we downloaded the DisGeNet genes-diseases metadata database of [38]. This provides information for grouping the genes by diseases they are associated with, making it useful for incorporation into the visualization of our learned GRNs. The user can select a specific disease and view all the associated genes. Conversely, the user can select a specific gene and view the disease associated with that gene as well as all the genes associated with that same disease (Fig 5).

**BN-specific tools for visualization and inference.**   The tools we developed specifically for BNs are for visualizing parameters and performing probabilistic inference. In our Gaussian BNs, the parameters are shown as an interactive plot of the marginal Gaussian distribution, where the user can zoom and hover the mouse over the plot to see the distribution values. The distribution shown is the marginal distribution, which is the expected distribution for the variable when we do not have any other information.

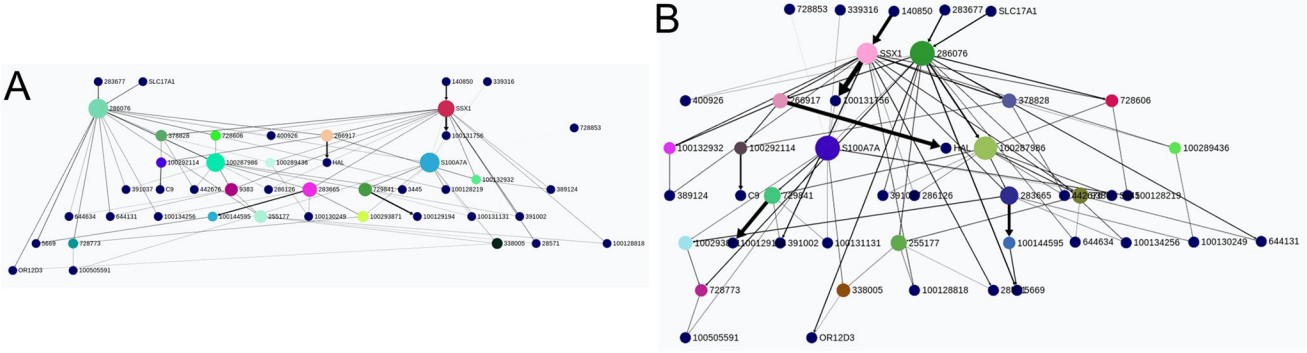

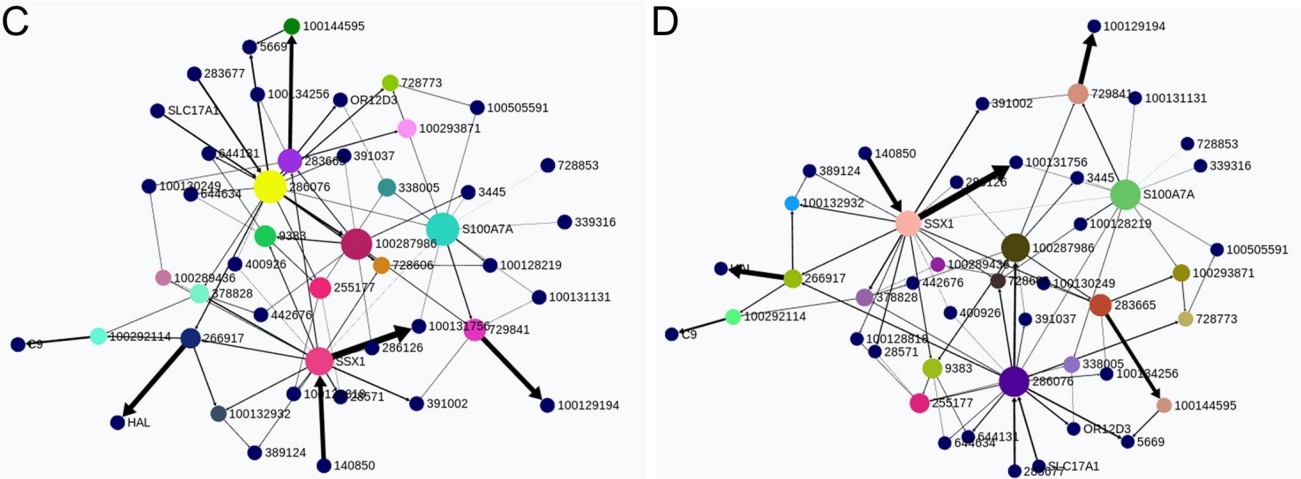

**Fig 4. Some layouts algorithms for BN visualization, using the same example network.** Note how the important nodes with the betweenness centrality algorithm are highlighted. **A**. Dot layout. **B**. Sugiyama layout. **C**. Fruchterman-Reingold layout. **D**. ForceAtlas2 layout.

To perform inference, the user must first set a specific value for a node or a group of nodes. This will set the selected nodes as evidence variables **E** and give them a special colour (red). The evidence variables, **E**, can be selected in three ways: selecting one specific node as evidence (Fig 6A), selecting a group of nodes as evidence (from an imported user defined groups file inspired by the Louvain algorithm) and defining a new list of nodes.

The inference is performed on the server side to provide an efficient implementation; therefore, we do not need the user to have a powerful computer that could help with usability. This allows us to run the inference in less than 30 seconds, even in a massive network with 20,000 nodes. The resulting joint distribution of all the variables before and after inference are cached in the backend to provide a faster visualization of the results. This means that the process is almost invisible to the user and is done only when evidence is fixed.

The second step is to visualize how the distributions of other nodes have changed with respect to the original ones before setting the evidence values. The user can either click on or search for a specific node or select a group of query nodes **Q** to obtain the posterior $p(\mathbf{Q}|\mathbf{E})$. This will be shown as an interactive Gaussian probability density function plot in black, while the original distribution will be shown in the same plot in blue (Fig 6B).

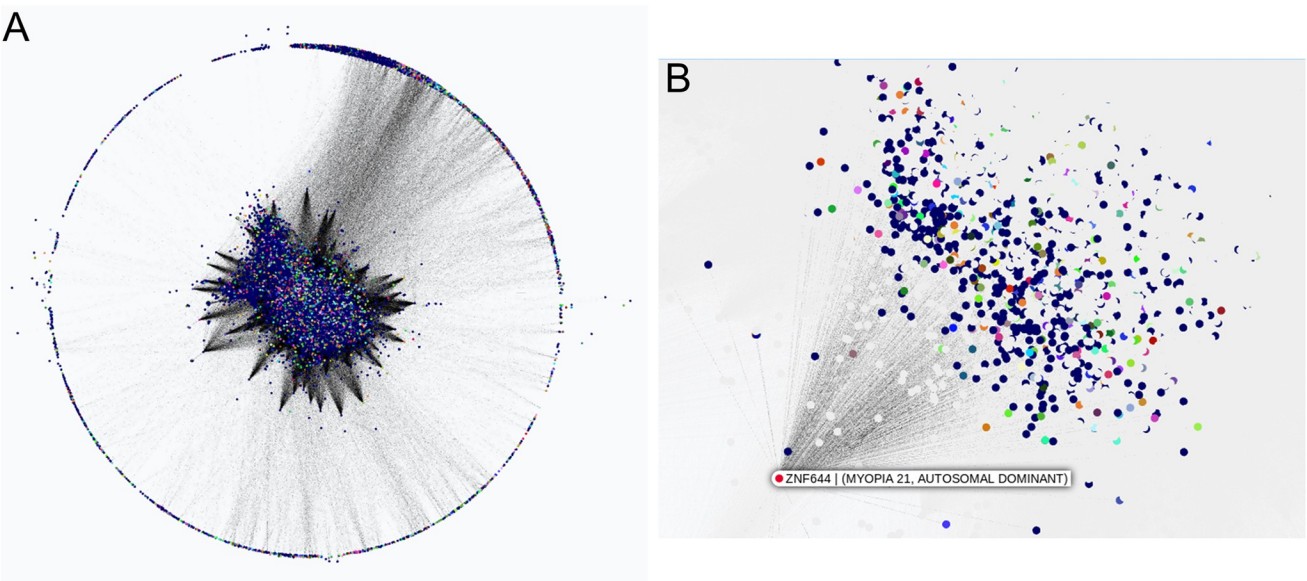

**Fig 5. Nodes colored by the genes-diseases metadata association from the DisGeNet database.** Our learned full brain human network. **A**. The color of each gene is determined by the main disease associated to it. **B**. Node ZNF644 selected to view its disease association (Myopia) and all its children genes related to it.

Finally, to support visual differential analysis, we created a tool that allows for comparison between two networks. This tool works by overlapping two networks with the same nodes so that the differences between the edges are shown with different colours, as in Fig 7. We also provide a tool for showing only the edges of the first network, the edges of the second network or the common edges.

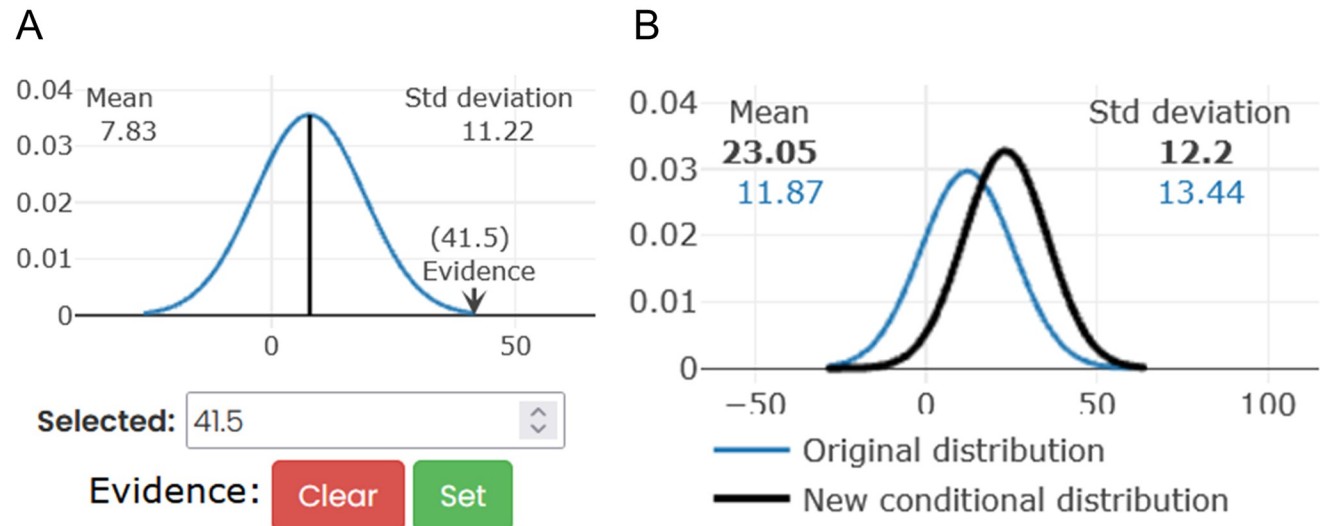

**Fig 6. Inference workflow in the BN visualization. A**. Set the evidence $E = e$ in a specific node. In this case, for a node with mean 7.83 and standard deviation 11.22, we set the observed value to 41.5. **B**. $p(Q|E = e)$, the posterior distribution of a node $Q$ given the evidence $E = e$. In blue, the marginal distribution of the node with no evidence and in black the distribution conditioned on the evidence.

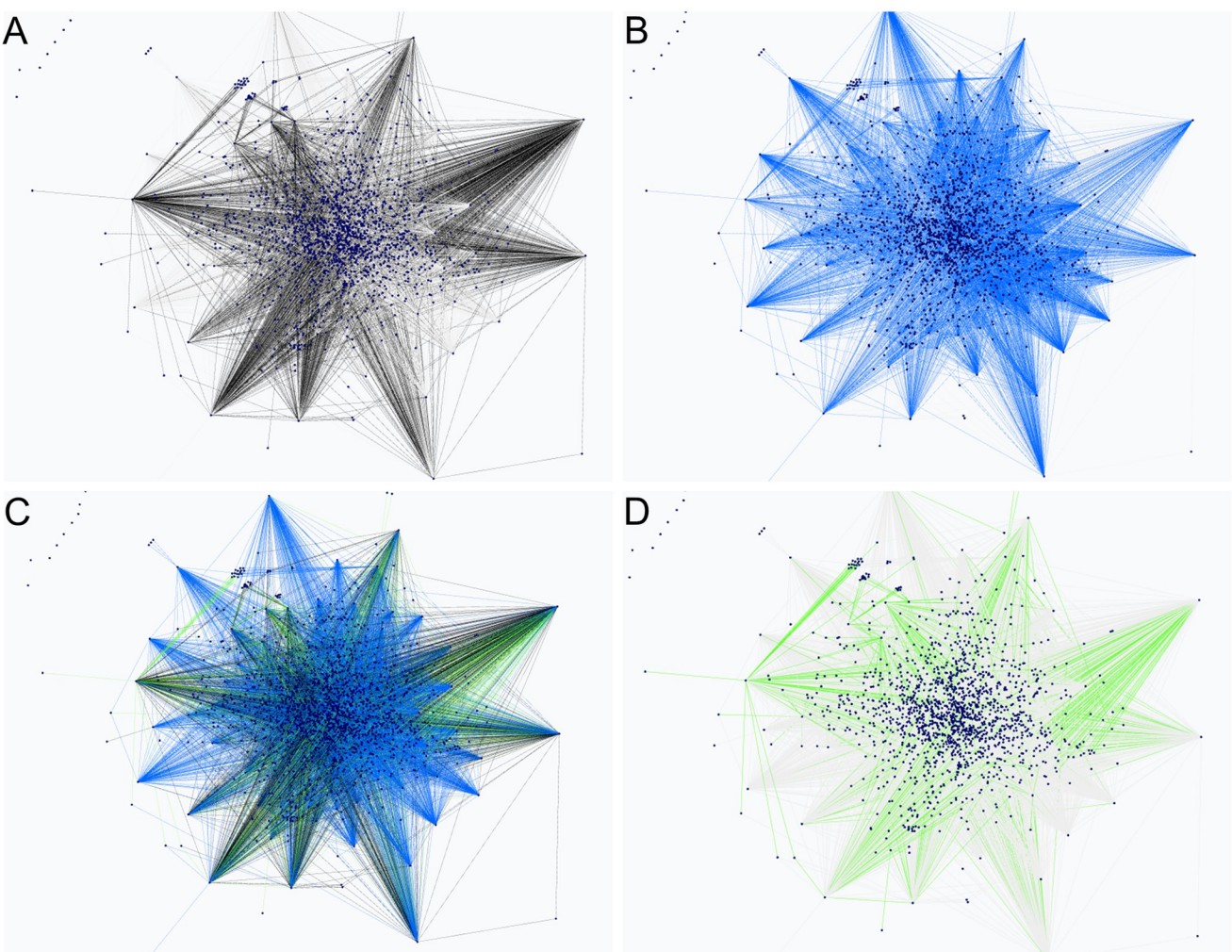

**Fig 7. Structure comparison of Network 1 of the Network Inference Challenge. A**. True graph edges. **B**. FGES-Merge learned graph edges. **C**. All edges. True graph edges are displayed in black, FGES-Merge learned graph edges in blue and common edges in green. **D**. Common edges between the true graph and our learned graph with FGES-Merge.

All the software has been packaged as a Docker [39] container to provide a production ready solution. Since the computationally costly code is running in our backend, the users only need an average GPU card to learn and visualize massive BNs fluently.

## Results

Our goal was to evaluate the performance of our method in recovering the underlying structure of GRNs compared to various algorithms, including both BN-based and non-BN-based methods. Additionally, we aimed to demonstrate that our algorithm overcomes common challenges in learning BNs and can scale to thousands of nodes.

To make these comparisons, we conducted two different tests. First, we assessed our accuracy in recovering the structure of the networks from the Network Inference Challenge [31], specifically in Network 1, 3, and 4, since Network 2 was excluded from the original challenge.

Second, we compared our method to other popular methods of learning BNs using the same data, focusing on the time required to learn the structure rather than the accuracy.

All experiments for both the benchmarks and final results were conducted in our MPI cluster with three nodes, each one running in Ubuntu 16.04, Intel i7-7700K CPU 4 cores at 4.2 GHz, and 64GB RAM.

### Structure recovery benchmark

We obtained the structure learned by all original competitors from the Network Inference Challenge from the results repository available at https://www.synapse.org/#!Synapse:syn2787211. This allowed us to compare our results with those of the competitors without having to run all the algorithms ourselves, which might have been infeasible due to the unavailability of some methods and the expected long runtime of others.

Each benchmark method outputs a matrix of $n \times n$ entries, each representing the probability assigned to one edge in the network. BN methods do not produce this output natively, and we believe that this might have been one of the reasons they performed so poorly during the original challenge. Our solution was to establish a threshold for all the methods and transform their probabilities into a series of binary predictions. To do this, we had to take into account that the network is sparse, so the prior probability for the binary classification problem is not 0.5 but instead approximately 1/number of nodes (from the discussion in the section on topological properties of GRNs).

The original score for the challenge was the area under the precision-recall curve (AUPRC), which is usually a good score for imbalanced problems. However, calculating the AUPRC properly for our method was challenging due to the absence of edge probabilities in BN methods. We could have estimated one by finding multiple structures for the same network while changing the hyperparameters of the model ($s_{max}, \lambda, \mu$). This proved to be computationally unfeasible since even though our method is much faster than other BN methods, the runtime for the smallest networks was still approximately 2 hours and was much higer for bigger networks. Estimating the AUPRC thusly would have massively increased the time required to run our experiments. In the end, we decided to use a score that has similar properties to AUPRC that we could compute much more easily, the Matthews correlation coefficient (MCC) [40], which is an extension of the F-score to deal with class imbalance. The expression for the MCC from the confusion matrix is:

$$MCC = \frac{TP \times TN - FP \times FN}{\sqrt{(TP + FP)(TP + FN)(TN + FP)(TN + FN)}} \tag{13}$$

where *TP, TN, FP*, and *FN* are true positives, true negatives, false positives and false negatives respectively. The MCC ranges from 0 to 1 with 0 meaning that the classifier has misclassified a whole class (predicting all edges or all non-edges) and 1 meaning perfect classification. Fig 8 shows the MCC scores for all the methods in the Network Inference challenge and FGES--Merge for Network 1 (*In-silico*, Fig 8A), 3 (*E. coli*, Fig 8B) and 4 (*Saccharomyces cerevisiae*, Fig 8C). FGES-Merge recovers the structure of the original network with scores comparable to the other BN methods (see Table 1 for a description of each method. For the other methods see Table 1 in [31]).

### Times benchmark

For our times comparison, we tested some of the most common BN learning algorithms implemented in the bnlearn R package [42] and one method based on tree ensembles (GENIE3) [43] on the smallest network ($\approx 1000$ nodes) from the Network Inference Challenge

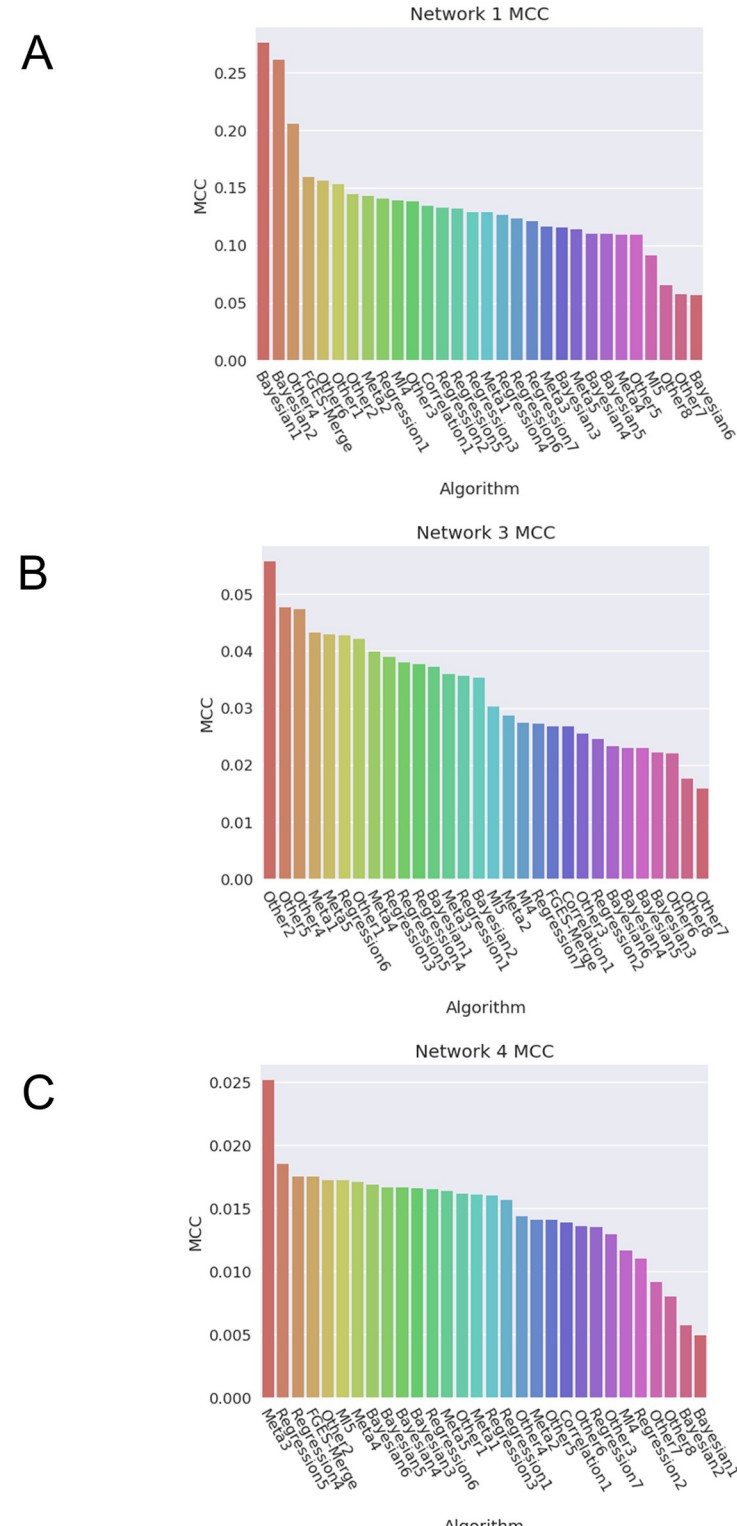

**Fig 8. MCC scores for all the methods in the Network Inference challenge and FGES-Merge. A**. Network 1 (*In-silico*). **B**. Network 3 (GRN for *Escherichia coli*). **C**. Network 4 (GRN for *Saccharomyces cerevisiae*). FGES-Merge comparable with other BN methods in **A** and **B**, and with the best non-BN methods in **A**. Furthermore, in **C** where methods 1 and 2 of BNs have very poor scores, FGES-Merge is still one of the best.

**Table 1. List of Bayesian network methods.**

| | Description |
|---|---|
| 1 | Simulated annealing (Catnet R package, https://cran.r-project.org/src/contrib/Archive/catnet/), aggregation of three runs. |
| 2 | Simulated annealing (Catnet R package, https://cran.r-project.org/src/contrib/Archive/catnet/). |
| 3 | Max-Min Parent and Children algorithm (MMPC), bootstrapped datasets [28]. |
| 4 | Markov blanket algorithm (HITON-PC), bootstrapped datasets [30]. |
| 5 | Markov boundary induction algorithm (TIE*), bootstrapped datasets [41]. |
| 6 | Models transcription factor perturbation data and time series using dynamic Bayesian networks (Infer.NET toolbox). |

(Network 1). We compared ourselves with GENIE3 because it perormed well in the challenge and the authors give a computational complexity that is log linear in the number of samples and at worst quadratic in the number of genes, comparable to the Chow-Liu algorithm, which has a complexity of $O(n^2)$, so we expected both of them to be faster than FGES-Merge. In the end, we could not test more than the smallest network since although our method finished in slightly more than an hour, the other non-quadratic BN methods ran for more than a day and did not converge.

The results present in Fig 9 show that our algorithm is slower than that of Chow-Liu, as expected, but slightly faster than GENIE3 which we did not expect. These results are surprising because they imply a massive improvement in speed between FGES-Merge and most other BN learning algorithms with very few restrictions.

## Human brain regulatory network

As part of the Human Brain Project, we have developed Neurogenpy, a Python library for learning and manipulating GRNs, and an associated plugin for siibra-explorer, the

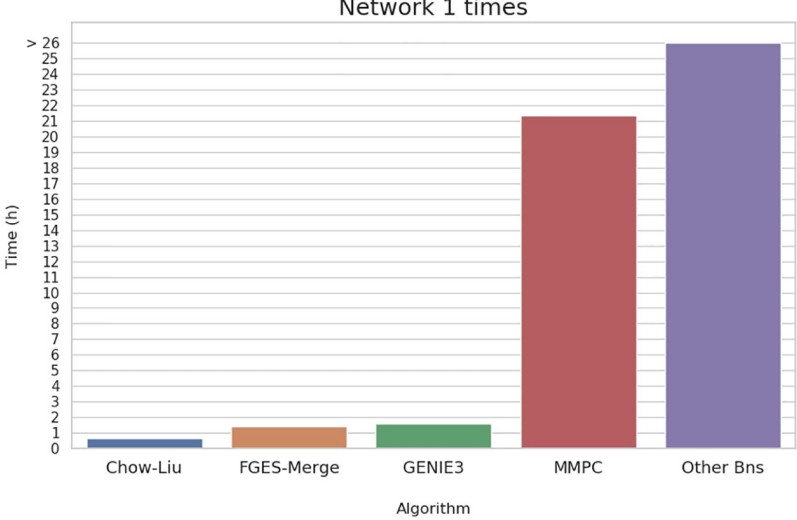

**Fig 9. Hours taken for different GRN learning methods for Network 1 of the Network Inference Challenge.** Among the BN learning methods only Chow-Liu's algorithm is competitive in time which is normal since its complexity is $O(n^2)$. However, GENIE3, a tree-ensemble based method with a complexity also bound by $O(n^2)$, is slower than FGES-Merge. Algorithms that did not finish before the 26 hour mark were forced to stop early.

visualization tool for the EBRAINS multilevel human brain atlas. FGES-Merge and BayeSuites have been developed in the context of these tools as ways to ensure that our methods both for learning and visualizing GRNs scale to any number of genes of interest from researchers using the atlas.

To showcase the ability of FGES-Merge to scale to thousands of nodes, we decided to apply our method to the Allen Brain Atlas gene expression dataset, which is the biggest gene expression dataset currently available in the EBRAINS atlas. We took all microarray data from the dataset and filtered it to only the protein coding genes (20,708 genes after filtering). Then, we used FGES-Merge to obtain a series of GRNs for some of the high-level anatomical structures defined by the ontology of the EBRAINS atlas.

In this section, we will show some of the generated networks, comment on possible applications of our models and discuss the topological properties of the learned networks to see if they respect the known empirical properties of GRNs.

**Networks learned.** Fig 10 shows two networks obtained with the whole dataset, i.e., GRNs using all the available samples for the whole brain. These GRNs were learned with different penalty parameters for FGES-Merge (see Eq 11) and thresholds for the number of edges. We can readily see how the higher-penalty network (Fig 10A) presents various disconnected components unlike the lower-penalty network (Fig 10B). This is what we would expect since a higher penalty forces sparsity. The second network predicted more edges than expected so the pruning step of FGES-Merge was used to remove some of the worst edges.

These networks can be used to visually search for relationships between genes, such as those shown in Fig 5, which can then be tested to obtain possibly useful information about diseases or development. It can also be used quantitatively as in Fig 6 to obtain concrete predictions about the effects of some genes on other genes' expression levels. This could complement clinical trials that aim to alter gene expression by helping researchers decide which genes might be good targets for medication and to explore possible side-effects.

Finally, multiple networks learnt by taking only the samples that come from different areas of the brain could be visualized as in Fig 7 to serve as an aid for differential analysis. Instead of observing only the gene expression levels between different conditions, we could see the structure and parameter changes, obtaining a clearer picture of the differences in gene regulation that lead to differences in gene expression.

**Topological properties of the learned GRNs.** Fig 11 shows the in-degree and out-degree distributions for the human brain GRN with fitted power-law and kernel density distributions for both. The in-degree distribution seems to follow the power-law distribution well, although with some unexpected gaps which might have to do with some artifacts from the cutoffs from the hyperparameters of FGES-Merge. The out-degree distribution is flatter in the middle but does seem to follow a curve at the tail. A better exploration of the topological behaviour of the networks would help understand when the assumptions we gave for the algorithm break.

The main reason to present these plots is to show that the algorithm succeeds in the task of reconstructing a network with neighbourhoods of thousands of nodes despite limiting the size of each subnetwork. Furthermore, the fact that the maximum size for the in-degree is less than the chosen $s_{max}$ makes us think that we overshot on the limit for the neighbourhoods, which would partly explain the unrealistc results obtained since we should not expect any regulator to have over a thousand children. We have to remember, however, that the found links between genes cannot be interpreted directly as regulatory links, since they are probabilistic relationships and it is easy to find false positive edges between nodes that are regulated by the same hub (specially since we made it more likely by using too big a size for the neighbourhood).

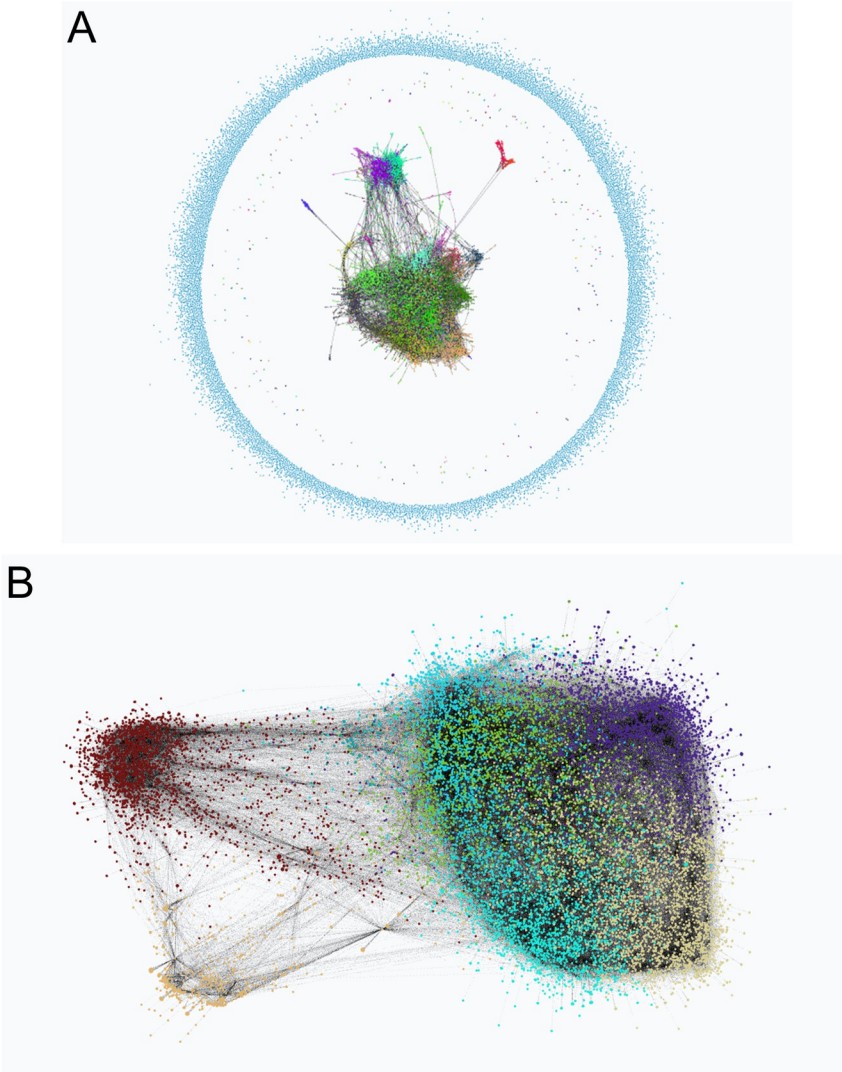

**Fig 10. Human brain genome networks.** The figure shows two networks learned with the whole Allen Brain Atlas dataset with different parameters for FGES-Merge. Both are visualized using BayeSuites and colored via the Louvain algorithm to identify groups of genes. Each color represents a community found by the algorithm. **A**. Network learned with a high penalty ($\lambda = 65$) for the FGES-Merge algorithm. **B**. Network learned with a lower penalty ($\lambda = 45$) for the FGES-Merge algorithm, both with $s_{max} = 100$.

## Discussion

### Limitations and further work

In this section, we discuss the limitations of the FGES-Merge method and acknowledge the potential trade-offs and simplifications made in our approach. We also present some possible avenues of further research on how FGES-Merge could be adapted to address these limitations.

One limitation of FGES-Merge is the assumption that gene expression data follows a linear Gaussian distribution, which might not hold in all cases [44]. While this assumption allows for computational tractability, specially of inference and parameter learning, it may introduce

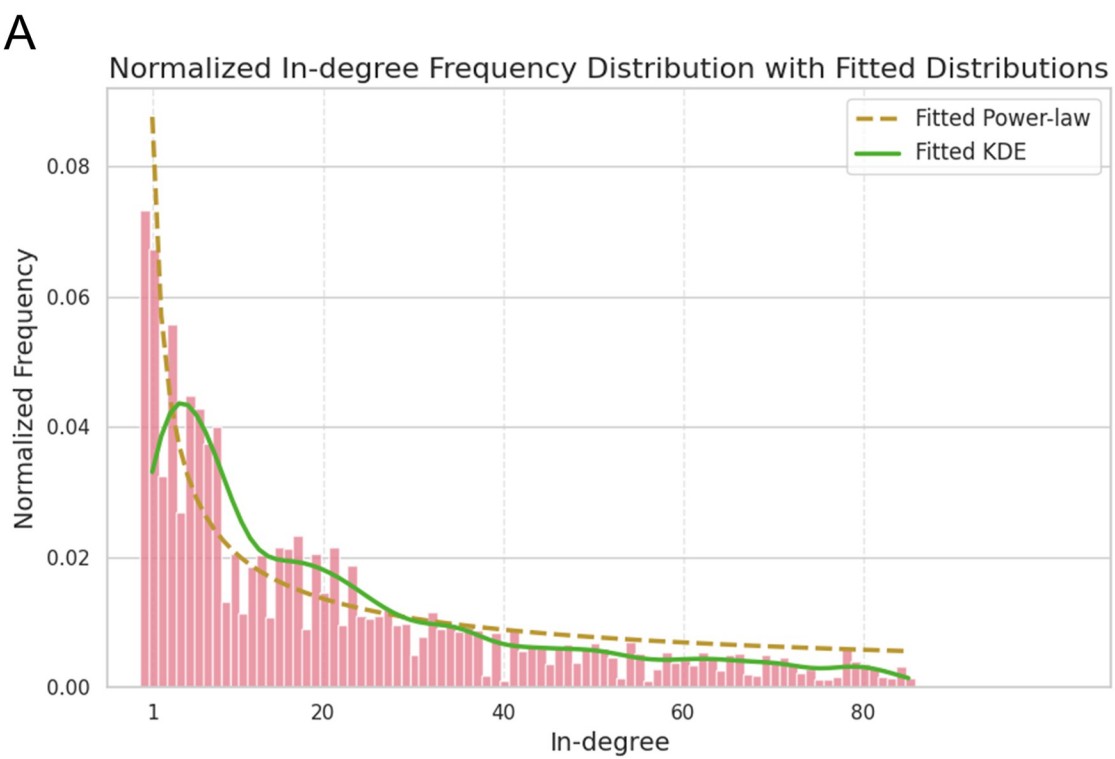

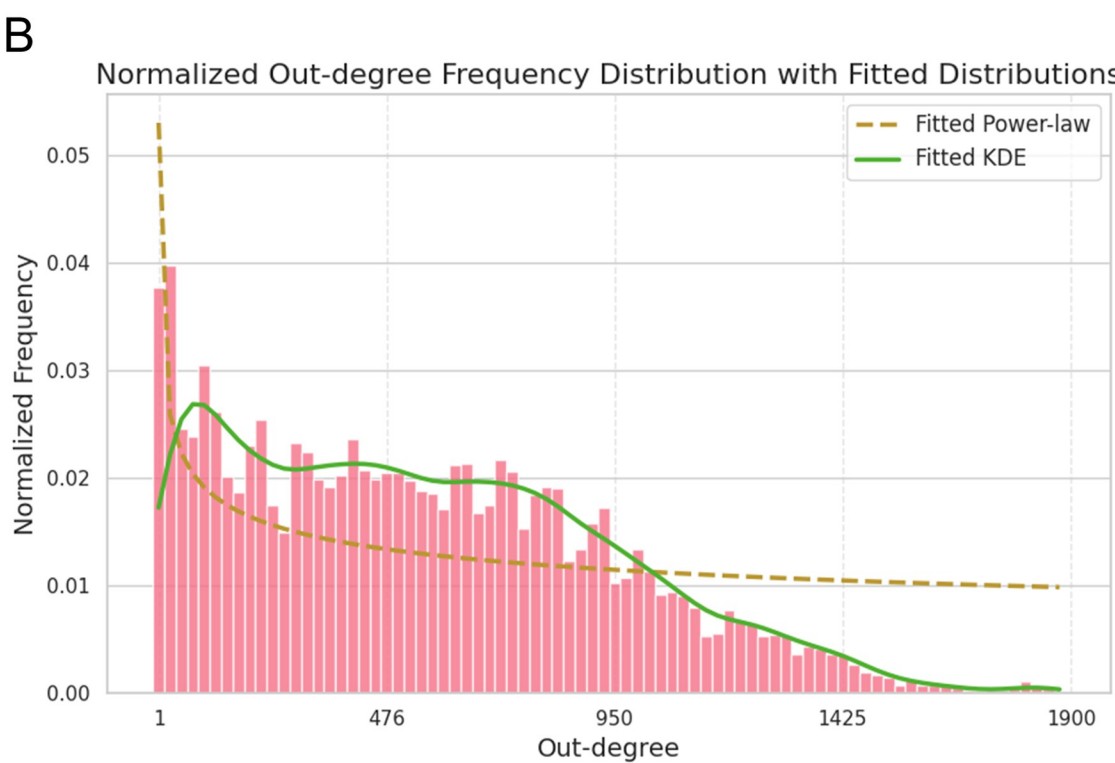

**Fig 11. Degree distributions for our learned full human brain GRN. A**. Node in-degree histogram. **B**. Node out-degree histogram. The network corresponds to Fig 10B.

inaccuracies in some situations. To address this limitation, future research could explore alternative distributions or non-parametric methods that can accommodate non-Gaussian data, trying to fit the proper distribution to each gene in the study [45]. This assumption is not essential to the current FGES-Merge implementation as long as a proper scoring function could be defined and tractably calculated for the edge additions (for example, such as in [46]), since that is the only place where the assumption is used in the algorithm. However, inference does become less tractable for a general distribution and even if the structure could be learnt in principle using a variant of FGES-Merge, we would need an inference method that could scale to the size of the network.

Another way to improve the ability of FGES-Merge to find relevant biological knowledge is by relaxing the assumption that all interactions have to be gene-to-gene. By considering heterogeneous data, including the concentration of other molecules in the cell, we could find relationships between them and gene regulation. This will be necessary if we want to have a more complete picture of the regulatory network.

Another clear limitation comes from assuming that gene regulation is static. Extending FGES-Merge to dynamic networks would require incorporating time-series data and taking into account time delays and feedback loops in the regulatory relationships. This is something for which we believe FGES-Merge is very well suited. A detailed explanation of dynamic BNs is out of the scope of this work, but given that considering temporal interactions increases the size of the search space for structures enormously (for a network of $n$ nodes, considering temporal interactions from up to $t$ temporal states, the number of edges grows as $O(n^2t!)$). Both adding dynamic data and non-genetic nodes make the number of nodes higher and clearly benefit from FGES-Merge scaling to massive networks.

Finally, the genome sized GRNs obtained by applying our method to the human brain using microarray data are not directly interpretable as encoding regulatory relationships between genes. This is because the data used is averaged over large tissue samples, so the data does not correspond to single cells and it is hard to give a direct biological interpretation of what the network represents. However, we believe the result is still interesting as a showcase of the ability of our algorithm to scale to massive networks using real data in the context in which it will mostly be used. Furthermore, the networks learnt from different anatomical structures allow us to showcase the capabilities of our visulization tool to compare networks of the same genes learnt in different contexts to find targets for research. This same method could be used with disease microrray data, comparing the resulting networks to healthy ones to try to find genes related to a given disease.

## Conclusion

We present FGES-Merge, an algorithm for finding the structure of a GRN from gene expression data that uses a divide-and-conquer approach to learning local BNs for each gene through parallelized heuristic search over the space of edge additions restricted by the topological properties of GRNs and combines them to form a BN over all the genes. The algorithm scales to tens of thousands of nodes, being able to deal with networks much bigger than the previous fastest methods using BNs while also being competitive with the state of the art in its ability to recover the underlying structure of the GRN as measured by the Matthews correlation coefficient.

We also present a web tool, BayeSuites, to be able to learn, visualize and manipulate massive BNs, including running inference from evidence and comparing the structure of pairs of networks. The algorithm is available as part of the Neurogenpy Python library at https://github.com/javiegal/neurogenpy, while BayeSuites is available at: https://neurosuites.com/morpho/

ml_bayesian_networks/. All the networks presented throughout the paper are available at: https://gitlab.com/mmichiels/fges_parallel_production/tree/master/BNs_results_paper.

Although the task of truly learning all regulatory interactions at the genome level is far from complete, we believe that FGES-Merge lifts a technical limitation on scaling BN learning to thousands of nodes in the GRN context and is a stepping stone for other methods that want to take it further through adding dynamic interactions or using more general distributions to model the expression data.

## Acknowledgments

The authors would like to thank Sergio Paniego and José Luis Moreno for their help in the development of BayeSuites and Javier Gallego for his work developing Neurogenpy and integrating the code for FGES-Merge into the Python library.

## Author Contributions

**Conceptualization:** Niko Bernaola.

**Data curation:** Niko Bernaola.

**Formal analysis:** Niko Bernaola.

**Funding acquisition:** Pedro Larrañaga, Concha Bielza.

**Investigation:** Niko Bernaola, Mario Michiels.

**Methodology:** Niko Bernaola, Mario Michiels.

**Project administration:** Pedro Larrañaga, Concha Bielza.

**Resources:** Pedro Larrañaga, Concha Bielza.

**Software:** Niko Bernaola, Mario Michiels.

**Supervision:** Pedro Larrañaga, Concha Bielza.

**Validation:** Niko Bernaola.

**Visualization:** Niko Bernaola.

**Writing – original draft:** Niko Bernaola, Mario Michiels.

**Writing – review & editing:** Niko Bernaola, Pedro Larrañaga, Concha Bielza.

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
