## [Decision Letter · Decision Letter 0]

20 Jan 2023

Dear Bernaola,

Thank you very much for submitting your manuscript "Learning massive interpretable gene regulatory networks of the human brain by merging Bayesian Networks" for consideration at PLOS Computational Biology.

As with all papers reviewed by the journal, your manuscript was reviewed by members of the editorial board and by several independent reviewers. In light of the reviews (below this email), we would like to invite the resubmission of a significantly-revised version that takes into account the reviewers' comments.

We cannot make any decision about publication until we have seen the revised manuscript and your response to the reviewers' comments. Your revised manuscript is also likely to be sent to reviewers for further evaluation.

Sincerely,

Mingyao Li

Academic Editor

PLOS Computational Biology

Lucy Houghton

Staff

PLOS Computational Biology

Reviewer's Responses to Questions

**Comments to the Authors:**

Reviewer #1: Bernaola et al. developed a novel method of mapping regulatory networks in the brain known as Fast Greedy Equivalence Search (FAGES), which merges locally learned Bayesian networks based on the fast greedy algorithm. They utilize microarray data from Allan brain atlas and compare benchmarks to the DREAM5 challenge. This is a very detailed and well written paper and the technology would be very useful for functional geneticists and researchers studying the brain. However there are some comments:

Major Comments:

1) Author’s associate KIF17 a gene associated with schizophrenia to KCNIP3 a gene associated with breast cancer. It is not clear if the microarray data used is geared to make this conclusion. KIF17 levels are likely changed by schizophrenia itself, were groups of patients with and without schizophrenia compared to make this conclusion. It is not clear from the study.

2) Might be worth while to look at unbias function of some the networks found. DAVID and other tools exist where lists of genes can be inputted and top functions are identified.

Minor:

1) Figures are hard to navigate. There is a lot of color used and it is difficult to interpret what they mean if any.

2) Discussions about how disease state microarray can be used in FAGES algorithm.

Reviewer #2: This work deals with the reconstruction of large gene regulatory networks (GRN) from gene expression data. The authors present an ad hoc algorithm to learn a large Bayesian network (BN),FGES-Merge; they motivate some choices, show results on simulated data and on a human 'brain' gene expression data set.

Main questions:

- it is not clear whether a 'genome-scale' GRN is truly informative. A GRN will be useful in deciphering molecular mechanisms during a given biological process (your subnets in the brain gene expression analysis). An overarching GRN is likely an utopia. Mechanisms are not activated all at once. Even if there was a one encompasses them all GRN, only subsets, possibly with different relationships (and some common OK) would be visible in clearly defined conditions. This needs to be clearly explained. For example, the use of 'averaged' gene expression to reconstruct the global GRN doesn't seem to make sense. Think of the local interaction between two genes that could be very different between two conditions (e.g. cancer or healthy patient). What would the 'average' represent?? Same for different brain regions.

- Avoid referring to the challenge data you use as the DREAM5 challenge data set. The latter was concerned with Genetical Genomics data (marker and expression data), see https://www.synapse.org/#!Synapse:syn2820440/wiki/71025. I agree that the Network Inference challenge you work on is coined DREAM5 by the Marbach et al. paper though.

- Some statements need at least a reference, see minor points below. More generally, introduce what you need and justify it bullet-proof, otherwise it's speculations. At best can be discussed. And your discussion section is very scarce. Actually more a quick conclusion.

Might repeat myself, but very little additional work, a clearer organisation of the argument, would convince your readership so much better, make that communication.presentation effort, your work is worth it!

Minor questions along the text:

The abstract is clear, thx

l5 'single-cell' -> will you use single-cell data? If not, remove, a general remark, simplify your work, do not get encyclopaedic, stick to what is of interest to *your* work.

l5 "most of the information about the development and function..." -> dangerous sentence. What about epigenetics? Genome plasticity?

l53 'However": what is the link between the 2 sentences?

l55 "the model what model?? Also check the reference's relevance on l56

l66: ref [14] doesn't have interactions

l77 so what does the measurement represent? How to compare samples?

l96 as you pointed out, a precise definition is key. Still waiting for yours at this stage

l129 no, this does not imply that the connected components (cc) are dense!

l133 a ref (local sparsity requirement)?

l142 definition of 'self-similar'?

l195 [27] is NOT "very" recent. Sometimes a general issue in your work, you rely on some old references in a very quickly moving field.

l204 conditional independence -> some algorithms ignore this (score-based)

l208-209 try and be more precise. Maybe even accurate.

l211 "output at each node" -> ??

l219 the [brain] is "not so well studied" -> so why using it for your showcase?? Not logical.

l253 computational cost an issue OK, more more fundamentally: the high-dimensional aspect of the problem. Needs to organise your thoughts on those issues, they appear here and there in the manuscript.

l255: first goes with second, firstly with secondly, sorry, being pedantic.

Clarify again the different nets for different regions (aka conditions) vs a unique global network.

l258 limitationS

Examples of 'other questions that might..."

l270 give the general concept of your novelty

l299 local BIC -> why this choice? Motivated later in the text only. Have you compared your approach to that using the MI? Also in Fig 1, refer to text when mentioning BIC (self-sufficient captions).

l305 -> well I would disagree, what about single-cell data you mention? More like small counts. Also in Appendix A "although an estimate can be found but with no guarantee of correctness" -> wrong, lots of research work has gone in this direction! Revise your arguments and make justice to people working on theoretical aspect of high-dimensional problems.

l309: what kind of inference? Probability queries? Where does this plug-in, it seems to come on top of an already very complex setting.

l310-315: order in the logical flow?

l351: this would be an ideal behaviour. What guarantee do we have this will occur?

Next line, rewrite X_i and X_j in math mode.

l356 "original" refers to? l357, isn't the argument upside-down?

l374 improved over [26]? Careful in re-using challenge data sets and performing better? It is considered as a sort of over-fitting since you know the results, hence optimise your method accordingly in a sense. E.g. one could argue you use the MCC criterion instead of the AUPR to show improved performance.

l391 Ref for equations in "Inference"?

Eqn (6) etc. What if E is not included in Pa(Y)? Seems clearer later, but was wondering what you were doing at this stage.

l444-45 a reference? Why show unsueful layouts in Fig. 4?

l454 a reference?

l458 does the data reflect this?

l468 parents observed at average value seems very crude, so no visualisation of interactions?

l478 'original joint distribution' yet another example of a conceptused before it is clarified.

How do you compare distributions? The Kullback-Leibler divergence is a classical ones. Others have been used...

l523 GG entries -> explain the notation

Fig. 7 (d) seems to indicate little overlap. Comments?

l536 How about varying the s_{max}'s?

l538 argument: well biologists could test according to the ranking of edge likelihood. And/or their prior interest in some genes/relationships/

l555 say more explicetly 1 and 2 refers to the BN methods labelled Bayesian1/2 in your plots

l562-564: very thin argument.

l568 GENIE3 is more a tree ensemble method. Yes relying on a regression setting, but so is yours in the end.

l589 not clear what is done to obtain this series of GRNs. Why not pool those, it seems to be what you announced. Then you could analyse those. No real validation, but from the global topology perspective. Any publication on link? Overlap with know data bases, e.g. STRING?

l595 See e.g. https://academic.oup.com/bioinformatics/article/25/3/417/244497, some work make a better use of different yet related conditions. Average gene expression (l603) seems VERY dubious.

l607-608 how is this pruning performed?

l627 give counts of other nodes. 80 parents is HUGE Is this realistic?? The average value is quite high too.

l637 the odes WITH PARENTS

l641 too dense? Also l642-643 are not convincing (me).

Would be good to have a discussion. Clearly missing. Sometimes, embryos are present in the results.

l648 yes the absolute scores are low. But what do they mean? If this gives enough new knowledge for future research, this is already a win. On the other hand, what if we hit a glass ceiling? Discuss!

l656 state-of-the-art

l663 queries such as? Not demonstrated in your work

Make sure the capital B is present in all references for Bayesian. E.g. [26] and [30]

**Have the authors made all data and (if applicable) computational code underlying the findings in their manuscript fully available?**

Reviewer #1: None

Reviewer #2: Yes

PLOS authors have the option to publish the peer review history of their article (what does this mean?). If published, this will include your full peer review and any attached files.

Reviewer #1: No

Reviewer #2: **Yes: **Matthieu Vignes
---

## [Decision Letter · Decision Letter 1]

19 Aug 2023

Dear Bernaola,

We are pleased to inform you that your manuscript 'Learning massive interpretable gene regulatory networks of the human brain by merging Bayesian Networks' has been provisionally accepted for publication in PLOS Computational Biology.

Best regards,

Mingyao Li

Academic Editor

PLOS Computational Biology

Lucy Houghton

%CORR_ED_EDITOR_ROLE%

PLOS Computational Biology

Reviewer's Responses to Questions

**Comments to the Authors:**

Reviewer #1: Paper has been restructured to focus more on technique rather than biological significance. As a methods paper this is acceptable and can be considered for acceptance.

**Have the authors made all data and (if applicable) computational code underlying the findings in their manuscript fully available?**

Reviewer #1: Yes

PLOS authors have the option to publish the peer review history of their article (what does this mean?). If published, this will include your full peer review and any attached files.

Reviewer #1: No

---

## [Editor Report · Acceptance letter]

27 Nov 2023

PCOMPBIOL-D-22-01339R1 

Learning massive interpretable gene regulatory networks of the human brain by merging Bayesian Networks

Dear Dr Larrañaga,

I am pleased to inform you that your manuscript has been formally accepted for publication in PLOS Computational Biology. Your manuscript is now with our production department and you will be notified of the publication date in due course.

With kind regards,

Lilla Horvath
